# Conserved and Noncanonical Activities of Two Histone H3K36 Methyltransferases Required for Insect-Pathogenic Lifestyle of *Beauveria bassiana*

**DOI:** 10.3390/jof7110956

**Published:** 2021-11-11

**Authors:** Kang Ren, Ya-Ni Mou, Sheng-Hua Ying, Ming-Guang Feng

**Affiliations:** MOE Laboratory of Biosystems Homeostasis & Protection, Collegeof Life Sciences, Zhejiang University, Hangzhou 310058, China; 11807120@zju.edu.cn (K.R.); 11907036@zju.edu.cn (Y.-N.M.); yingsh@zju.edu.cn (S.-H.Y.)

**Keywords:** entomopathogenic fungi, histone lysine-specific methyltransferases, gene expression and regulation, asexual development, stress response, virulence

## Abstract

Set2 and Ash1 are histone methyltransferases (KMTs) in the KMT3 family normally used to catalyze methylation of histone H3K36 (H3K36me) but remain unexplored in fungal insect pathogens. Here, we report broader/greater roles of Set2 and Ash1 in mono-/di-/trimethylation (me1/me2/me3) of H3K4 than of H3K36 in *Beauveria bassiana* and function similarly to Set1/KMT2, which has been reported to catalyze H3K4me3 as an epigenetic mark of *cre1* (carbon catabolite repressor) to upregulate the classes I and II hydrophobin genes *hyd1* and *hyd2* required for conidial hydrophobicity and adherence to insect cuticle. H3K4me3 was more attenuated than H3K36me3 in the absence of *set2* (72% versus 67%) or *ash1* (92% versus 12%), leading to sharply repressed or nearly abolished expression of *cre1*, *hyd1* and *hyd2*, as well as reduced hydrophobicity. Consequently, the delta-*set2* and delta-*ash1* mutants were differentially compromised in radial growth on various media or under different stresses, aerial conidiation under normal culture conditions, virulence, and cellular events crucial for normal cuticle infection and hemocoel colonization, accompanied by transcriptional repression of subsets of genes involved in or required for asexual development and multiple stress responses. These findings unravel novel roles of Set2 and Ash1 in the co-catalysis of usually Set1-reliant H3K4me3 required for fungal insect-pathogenic lifestyle.

## 1. Introduction

Signaling and epigenetic networks regulate cellular processes and events of filamentous fungal adaptation to both the host and environment. Understanding the regulatory role is of special importance for in-depth insight into the insect-pathogenic lifestyle of hypocrealean fungi, which serve as main sources of fungal insecticides [1]. Such fungi start host infection by conidial adhesion to insect integument, followed by germination, hyphal invasion into the insect body via cuticular penetration, and proliferation in vivo by yeast-like budding until host mummification to death. Upon host death, intrahemocoel hyphae penetrate the host cuticle again for outgrowth and produce on cadaver surfaces aerial conidia to initiate a new infection and/or disperse in the host habitats. Thus, the potential of fungal biocontrol against insect pests is an overall output of all cellular processes and events associated with normal infection, hemocoel colonization, responses to stresses outside and inside host, and asexual developmental [2,3,4,5].

The methylation of histone H3 (H3me) at specific lysine residues, such as K4, K9 and K36, is one of fundamental posttranslational modification mechanisms underlying DNA-based cellular events and RNA processing in eukaryotes [6,7,8] and relies upon histone lysine methyltransferases (KMTs) enabling to transfer one, two or three methyl groups to specific lysines for mono-, di- and trimethylation (me1/me2/me3). Such KMTs share a domain known as SET [Su(var)3–9, Enhancer-of-zeste and Trithorax] [9] and fall into the KMT1, KMT2 and KMT3 families [10]. In *Saccharomyces cerevisiae*, H3K4me is catalyzed by the H3K4 methyltransferase COMPASS (COMplex of Proteins Associated with SET1) consisting of KMT2/Set1 and protein partners [11], acts as epigenetic marks of actively transcribed genes [6,12], and is mediated by COMPASS-binding CxxC zinc finger protein 1 (Cfp1) [13]. The binding of COMPASS to mono-ubiquitinated histone H2B and free nucleosome [14] can define the transcriptional status of the genomic region and protect the genome from the replication stress and instability induced by transcription-replication conflicts [15]. H3K9me is catalyzed by the KMT1 enzymes associated with the silent regions of euchromatin and heterochromatin [9,16,17,18], such as Su(var)3–9 proteins in *Drosophila* [19,20], SUV39H1 and SUV39H2 in mammals [18] and cryptic loci regulator 4 (Clr4) in fission yeast [21]. H3K36me depends on the catalytic role of KMT3/Set2 [7,8]. Set2 is the unique KMT3 required for H3K36me in *S. cerevisiae* [22], contrasting with eight KMT3 enzymes to catalyze H3K36me in humans [23]. H3K4me3- or H3K36me3-marked nucleosomes are associated with transcription-active genes [24]. Set1 and Set2 mediate alternative polyadenylation at the cleaving sites of precursor mRNA [25].

Despite the intensive studies in model yeast, catalytic activities and biological effects of KMT1, KMT2 and KMT3 enzymes in filamentous fungal pathogens have been explored only in recent years. In *Botrytis cinerea*, H3K9me3 was nearly abolished due to loss-of-function mutation of Dim5 (BcDIM5) orthologous to Clr4 (KMT1) in fission yeast, leading to reduced virulence and marked defects in hyphal growth, conidiation and sclerotia formation [26]. Likewise, attenuated H3K9me3 led to defects in conidiation and perithecium production, reduced virulence, increased osmotolerance and hyper-phos- phorylated Hog1 when Dim5 lost function in *Fusarium verticillioides* [27]. Inactivation of H3K4me-required Set1 (KMT2) in *F. verticillioides* resulted in retarded growth, reduced virulence, blocked fumonsin B1 biosynthesis and repressed expression of phenotype- related genes [28]. Set1was shown to orchestrate the transcription of genes involved in secondary metabolism and act as a conidiation activator in *Fusarium*
*fujikuroi* [29]. The components MoBre2 (Cps60/ASH2L), MoSPP1 (Cps40/Cfp1) and MoSwd2 (Cps35) identified from the SET1/COMPASS of *Magnaporthe* (*Pyricularia*) *oryzae* were reported to regulate invasive hyphal development, pathogenicity and H3K4me-marked genes [30]. In *F. verticillioides*, H3K36me3-dependent Set2 (KMT3) was shown to mediate hyphal growth, pathogenicity, and secondary metabolism [31]. In *F. fujikuroi*, Set2 and Ash1, the other SET domain-containing KMT3 enzyme, were both characterized as mediators of H3K36me3 essential for vegetative growth, sporulation, secondary metabolism, and virulence [32], although a subsequent study revealed no role for an Ash1-like protein in methylating four H3 lysine residues including K36 in *M. oryzae* [33]. These studies demonstrate the conserved activities of KMT2, KMT1 and KMT3 enzymes in the respective catalysis of H3K4me, H3K9me and H3K36me and their important roles in the lifecycles in vitro and in vivo of plant-pathogenic fungi. However, the previous studies paid little attention to noncanonical activities of KMTs that may affect fungal biology and physiology. As an interesting example, the role of Set1 in hyphal growth, conidiation, sclerotia formation, aflatoxin B1 biosynthesis and virulence of *Aspergillus flavus* was evidently associated with the conserved activity of H3K4me2/me3 and also with the noncanonical activity of H3K36me2 [34], suggesting that some KMTs may methylate noncanonical H3 lysine residues as possible epigenetic marks of genes involved in filamentous fungal fitness to host and environment.

More recent studies have revealed catalytic activities and biological effects of major KMTs in insect-pathogenic fungi. In Hypocreales, the *Metarhizium* lineage in Clavicipitaceae is considered to have evolved insect pathogenicity from plant-pathogenic or endophytic fungi ~130 million years later than the *Beauveria*/*Cordyceps* lineage in Cordycipitaceae [1], resulting in some mechanistic differences of the two lineages in host infection and environmental adaptation [2,3,4,5,35]. In *Metarhizium*
*robertsii,* KMT2-catalyzed H3K4me1/me2/me3 was nearly abolished in the absence of *kmt2* (*set1*), leading to blocked appressorial formation, reduced virulence, and dysregulation of 1498 genes [36]. In the study, H3K4me3 acted as the preferential mark, and hence activated the expression, of *cre1*, a coding gene of carbon catabolite repressor 1 (transcription factor), and the activated *cre1* enabled the upregulation of the key hydrophobin gene *hyd4* to mediate the formation of appressoria required for host infection, thereby unraveling a regulatory role of the KMT2-Cre1-Hyd4 pathway in the fungal pathogenesis [36]. Indeed, *hyd4* in *M. robertsii* is homologous to both *hyd1*and *hyd2* encoding classes I and II hydrophobins in *Beauvera*
*bassiana*, in which *hyd1* and *hyd2* regulate hydrophobin biosynthesis and assembly into an outermost rodlet-bundle layer of conidial coat determinant to conidial hydrophobicity and adherence to insect cuticle [37]. Among all insect mycopathogens known to date, *B. bassiana* has the broadest host spectrum but does not form appressoria during hyphal invasion into the insect body [38] perhaps due to more powerful mechanisms that have evolved to infect diverse hosts. In *B. bassiana*, abolished H3K4me1/ me2/me3 and attenuated H3K36me2 concurred in the absence of *set1*, leading to the nearly or completely abolished expression of *cre1*, *hyd1* and *hyd2*, sharp repression of *brlA* and *abaA* as key activator genes of central development pathway (CPD), and severe defects in conidiation, host infection, proliferation in vivo and virulence [39]. The latter study unveils conserved and noncanonial activities of Set1 to H3K4me and H3K36me, respectively, and also the Set1-Cre1-Hyd1/2 pathway that functions in *B. bassiana* as does the KMT2-Cre1-Hyd4 pathway in *M. robertsii* and mediates not only cellular events associated with host infection and virulence, but also asexual development required for fungal survival/dispersal in host habitats. Likewise, Dim5/KMT1 showed conserved activities to H3K9me1/m2/me3 and broader noncanoical activities to H3K4me1/me2 and H3K36me2 and acted as a regulator of 1201 genes involved in cellular processes and events required for or associated with the lifecycle in vivo and in vitro of *B. bassiana* [40]. These studies uncover pleiotropic effects of Dim5/KMT/ and Set1/KMT2 on fungal insect- pathogenic lifestyle. However, KMT3 enzymes remain unexplored in fungal insect pathogens. This study seeks to characterize catalytic activities and biological effects of *B. bassiana* Set2 and Ash2, two KMT3 enzymes reported to catalyze H3K36me2/me3 in *A**. flavus* [34]. Unexpectedly, Set2 and Ash1 displayed broader/greater roles in the catalysis of H3K4me than of H3K36me and also the biological effects on the fungal virulence and related cellular events as Set1/KMT2 elucidated previously [39], as presented below.

## 2. Materials and Methods

### 2.1. Bioinformatic Analysis

The amino acid sequence ofSet2from *S. cerevisiae* S288C (NCBI accession code: NP_012367) was used as a query to search through the NCBI databases of *B. bassiana* and representative ascomycetes, including entomopathogens and nonentomopathogens, at https://blast.ncbi.nlm.nih.gov/Blast.cgi/ (accessed on 13 October 2021). Conserved domains and nuclear localization signal (NLS) motif were predicted from the Set2 homologs of four selected fungi at http://smart.embl-heidelberg.de/ (accessed on 13 October 2021) and http://nls-mapper.iab.keio.ac.jp/ (accessed on 13 October 2021), respectively. Sequence alignments were performed at http://www.bio-soft.net/format/DNAMAN.htm/ (accessed on 13 October 2021), followed by phylogenetic analysis with the maximum likelihood method in MEGA7 software at http://www.megasoftware.net/ (accessed on 13 October 2021).

### 2.2. Subcellular Localization of Set2 and Ash1 in B. bassiana

The plasmid pAN52-C-gfp-bar (C: 5′-*Pme*I-*Spe*I-*Eco*RV-*Eco*RI-*Bam*HI-3′ controlled by the homologous promoter P*tef1*) was used as a backbone to transform the fusion genes *set2-gfp* and *ash1-gfp* into the wild-type strain *B. bassiana* ARSEF 2860 (designated WT), as described previously [40,41]. Briefly, open reading frame (ORF) of *set2* or *ash1* was amplified from the WT cDNA with paired primers (Appendix A) and fused to the N-terminus of the green fluorescence protein gene *gfp* (GenBank: U55763) in the plasmid linearized with *Xma*I/*Bam*HI. The resultant plasmids were separately transformed into the WT genome via *Agrobacterium*-mediated transformation. Putative transformants were screened by the *bar* resistance to phosphinothricin (200 μg/mL). A transgenic strain from each transformation was selected based on its desirable green fluorescence signal and incubated for conidiation on SDAY (4% glucose, 1% peptone and 1.5% agar plus 1% yeast extract) at the optimal regime of 25 °C in a light/dark (L:D) cycle of 12:12 h. Conidia collected from the culture were suspended in SDBY (i.e., agar-free SDAY), followed by a 2-day incubation on a shaking bed (150 rpm) at 25 °C. Culture samples were stained with the nuclear dye DAPI (4′,6′-diamidine-2′-phenylindole dihydrochloride; Sigma-Aldrich, Shanghai, China), followed by laser scanning confocal microscopic (LSCM) analysis to determine subcellular localization of expressed Set2-GFP or Ash1-GFP fusion protein.

### 2.3. Construction of set2 and ash1 Mutants

The *set2* or *ash1* gene was disrupted by deleting a partial promoter/coding fragment of 467 or 574 bp (Appendix A) through homologous recombination of 5′ flanking and 3′ coding fragments separated by the *bar* marker of the deletion plasmid p038-5′*x*-bar-3′*x* (*x* = *set2* or *ash1*) in the WT strain and complemented into an identified Δ*set2* or Δ*ash1* mutant through the ectopic integration of a cassette comprising *sur* marker and a full-length coding sequence of each gene with flank regions in the complementation plasmid p0380-sur-*x*, as described previously [40,41]. The constructed plasmids were integrated into the WT and Δ*set2* or Δ*ash1* strains, respectively, as aforementioned. Putative mutant colonies were screened by the *bar* resistance to phosphinothricin (200 μg/mL) or the *sur* resistance to chlorimuron ethyl (10 μg/mL). Expected recombination events in the genomic DNAs of those mutants were examined via PCR (Appendix A) and real-time quantitative PCR (qPCR) analyses. Paired primers used for manipulation of each target gene are listed in Appendix A. The abolition of the Δ*set2* and Δ*ash1* mutants with targeted gene expression and the restoration of the Δ*set2::Set2* and Δ*ash1::Ash1* mutants with targeted gene expression to the WT level (Appendix A) were evaluated in parallel with the parental WT in experiments, each comprising three independent replicates per strain.

### 2.4. Western Blot for Catalytic Activity of Lysine-Specific H3me

Our previous protocols [39,40] were used to characterize lysine-specific H3me in the nuclear protein extracts of the deletion mutants and control (WT and complementation) strains in western blot experiments. Briefly, nuclear protein extracts were isolated from the 3-day-old cultures of 50 mL 10^6^ conidia/mL suspension in SDBY as described in the user’s guide of Nuclear and Cytoplasmic Protein Extraction Kit (Beyotime, Shanghai, China; Catalog No.: P0027). Aliquots of 40 μg protein extracts were loaded onto 12% SDS-PAGE, transferred to polyvinylidene difluoride (PVDF) membranes (Merck Millipore, Darmstadt, Germany), and probed for the signals of accumulated H3 and of mono-, di- and trimethylated H3K4, H3K9 and H3K36 with 1000-fold dilutions of the corresponding anti-methyl antibodies listed in Appendix A. The bound antibodies reacted with 5000-fold dilution of horseradish peroxidase (HRP) conjugated Affini Pure Goat Anti-Rabbit IgG (H + L) antibodies (Boster, Wuhan, China; Catalog No.: BA1054) and visualized in a chemiluminescence detection system (Amersham Biosciences, Shanghai, China). Three technical replicates were included in each western experiment. Signal intensities of all blots were quantified using the software ImageJ (https://imagej.nih.gov/ij/, accessed on 13 October 2021). The signal level of each methylated lysine residue relative to nuclear H3 accumulation was computed as the ratio of the methylated lysine intensity to the H3 intensity (H3Kme/H3 ratio).

### 2.5. Assays for Growth Rate, Conidiation Capacity, Conidial Quality and Stress Tolerance

Aliquots of 1 μL × 10^6^ conidia/mL suspension were spotted on the plates of rich SDAY, minimal Czapek-Dox agar (CDA: 3% sucrose, 0.3% NaNO_3_, 0.1% K_2_HPO_4_, 0.05% KCl, 0.05% MgSO_4_, 0.001% FeSOa_4_ and 1.5% agar) and CDAs amended with different carbon (glucose, trehalose, glycerol, and sodium acetate) or nitrogen (NaNO_2_, NH_4_Cl and NH_4_NO_3_) sources or with a deleted carbon or nitrogen source (starving). After an 8-day incubation at 25 °C and L:D 12:12, the diameter of each colony was estimated as a growth index using two measurements taken perpendicular to each other across the center.

Cultures used for measurements of conidial yields and biomass accumulation were initiated by spreading 100 μL aliquots of a 10^7^ conidia/mL suspension on SDAY plates (9 cm diameter) overlaid with or without cellophane, followed by a 9-day incubation at the optimal regime of 25 °C and L:D 12:12. From day 5 onwards, 3 samples were taken at a 2-day interval from each plate culture using a cork borer (5 mm diameter). Conidia in each sample were released into 1 mL of aqueous 0.02% Tween 80 by 10 min supersonic vibration. Conidial concentration of the suspension was assessed using a hemocytometer and converted to the number of conidia per unit area (cm^2^) of plate culture. The cultures from the cellophane-overlaid plates were dried for 3 h at 70 °C for quantification of biomass level the day before assessment of conidial yield. In addition, the quality of conidia from the SDAY cultures of each strain was evaluated using previous indices [40,41], including GT_50_ (h) as a viability index for 50% germination at 25 °C, hydrophobicity index assessed in an aqueous-organic system, and median lethal dose (LD_50_, J/cm^2^) for conidial resistance to UVB irradiation (weighted wavelength: 312 nm).

Stress assays were carried out by initiating colony growth as aforementioned on the plates of CDA alone (control) or supplemented with NaCl (0.8 M) or sorbitol (1 M) for hyperosmotic stress, menadione (0.02 mM) or H_2_O_2_ (2 mM) for oxidative stress, and Congo red (3 μg/mL) or calcofluor white (10 μg/mL) for cell wall perturbing stress, respectively. The diameter of each colony was measured as aforementioned after an 8-day incubation at 25 °C. Relative growth inhibition (RGI) was estimated as (*d*_c_ − *d*_s_)/*d*_c_ × 100 (*d*_c_ and *d*_s_: diameters of control and stressed colonies respectively) to reveal the sensitivity of each strain to each chemical stressor. In addition, SOD Activity Assay Kit (Sigma-Aldrich, St. Louis, MO, USA) and Catalase Activity Assays Kit (Jiancheng Biotech, Nanjing, China) were used to assess total activities (U/mg) of superoxide dismutases (SOD) and catalases, respectively, in the protein extracts of 3-day-old SDAY cultures following the manufacturers’ guides.

### 2.6. Bioassays for Fungal Virulence

The virulence of each fungal strain was assayed on the third-instar larvae of *Galleria mellonella* through two infection modes. Briefly, 3 groups of ~35 larvae were immersed for 10 s in 40 mL aliquots of a 10^7^ conidia/mL suspension for normal cuticle infection (NCI); 5 μL of a 10^5^ conidia/mL suspension was injected into the hemocoel of each larva in each group for cuticle-bypassing infection (CBI). All treated groups were maintained at 25 °C, and their survival/mortality records were noted every 12 h until no more change in mortality. Modeling analysis of the time-mortality trend in each group was performed to estimate median lethal time (LT_50_) as an index of virulence in either infection mode.

### 2.7. Examination of Cellular Events Crucial for Host Infection and Virulence

Cellular events associated with virulence via NCI or CBI were examined or analyzed as described previously [40,41]. First, conidial adherence essential for NCI initiation was assayed on locust hind wings as described elsewhere [38]. Briefly, conidia were suspended in sterile water (free of any surfactant to affect conidial surface trait) by thorough vortex and standardized to 10^7^ conidia/mL. Aliquots of 5 μL conidial suspension were spotted on the central areas of hind wings attached to 0.7% water agar and incubated for 8 h at 25 °C. Counts of conidia were made from 3 microscopic fields of each wing immediately after the incubation and made again after a 30 s washing of less adhesive conidia in sterile water. Percent ratios of post-wash versus pre-wash counts were computed as relative conidial adherence of each strain to the wing cuticle with respect to the WT standard. Second, larvae died from mycosis were incubated at 25 °C to observe whether intrahemocoel hyphae could penetrate insect cuticle for outgrowths on cadaver surfaces, revealing a capability of hyphal invasion into insect body via NCI. Third, 50 mL aliquots of a 10^6^ conidia/mL suspension in CDB (i.e., agar-free CDA) amended with 0.3% bovine serum albumin (BSA) as sole nitrogen source and enzyme inducer were incubated at 25 °C for 3 days on the shaking bed. The total activities (U/mL supernatant) of the extracellular (proteolytic, chitinolytic and lipolytic) enzymes (ECEs) and the subtilisin-like Pr1 family proteases, which are required for cuticle penetration and successful NCI [42,43], and biomass level (mg/mL) were quantified from each of the CDB-BSA cultures as described previously [43,44]. Fourth, hemolymph samples were taken from the larvae surviving 72 and 96 h post-CBI and microscopically examined for the presence and abundance of hyphal bodies (i.e., blastospores), which indicate a status of proliferation in vivo determinant to the speeds of mycosis development and host death. The concentration of hyphal bodies in each of nine samples from three larvae (three samples per larva) was assessed with a hemocytometer. Finally, 50 mL aliquots of a 10^6^ conidia/mL suspension in trehalose-peptone broth (TPB), a medium amended from CDB with the sole carbon source of 3% trehalose and the sole nitrogen source of 0.5% peptone to mimic insect hemolymph, were incubated at 25 °C for 3 days on the shaking bed. Estimates of blastospore concentration and biomass level in each of the cultures were made to compute dimorphic transition rate (no. blastospores/mg biomass) indicative of a capability of hemocoel colonization.

### 2.8. Transcriptional Profiling

To verify recombination events in the *set2* and *ash1* mutants, cultures were initiated by spreading 100 μL aliquots of a 10^7^ conidia/mL suspension on cellophane-overlaid SDAY plates and shaking 50 mL aliquots of a 10^6^ conidia/mL suspension in TPB, respectively, followed by a 3-day incubation at 25 °C. To assess the transcript levels of 32 phenotype-related genes in the mutants relative to WT, 3-, 4- and 5-day-old SDAY cultures were prepared as aforementioned. Total RNA was extracted from each of the cultures with an RNAiso Plus Kit (TaKaRa, Dalian, China) and reversely transcribed into cDNA with a PrimeScript RT reagent kit (TaKaRa). Transcripts of each gene were quantified from three cDNA samples derived from independent cultures via qPCR with paired primers (Appendix A) under the action of SYBR Premix *Ex Taq* (TaKaRa). The coding gene of glyceraldehyde-3-phosphate dehydrogenase (*GADPH*) was used as a reference. Relative transcript levels of *set2* and *ash1* in the SDAY and TPB cultures and of phenotype-related genes in the SDAY cultures of each mutant were computed with respect to the WT standard using a threshold-cycle (2^−^^ΔΔCT^) method. One-fold transcript change was used as a significant level. The phenotype-related genes included the CDP activator genes *brlA*, *abaA* and *wetA* and downstream *vosA* collectively required for conidiation and conidial maturation [45,46], five SOD (*sod1*–*5*) and six catalase (*cat1*–*6*) genes involved in the decomposition of superoxide anions and H_2_O_2_ [47,48], and the mitogen-activated protein kinase (MAPK) Slt2- and Hog1-cascaded kinase genes [49,50] and those likely associated with cell wall composition or integrity.

### 2.9. Statistical Analysis

All data from the experiments of three independent replicates were subjected to one-factor (strains) analysis of variance, followed by Tukey’s honestly significant difference (HSD) test for phenotypic differences among the tested strains.

## 3. Results

### 3.1. Domain Architecture and Phylogenetic Links of fungal Set2 and Ash1 Homologs

The BLASTp search with the yeast Set2 query resulted in recognition of two homologs from filamentous species of ascomycetes. The two homologs are Set2 and Ash1, which were characterized as H3K36me-specific KMT3 enzymes in *F. fujikuroi* [32], and fall into distinct clades in phylogeny (Appendix A). The *B. bassiana* Set2 (EJP64008, 900 aa, 100.9 kDa) and Ash1 (EJP63307, 824 aa, 89.78 kDa) are encoded by nucleotide sequences of 2877 bp with 2 introns and 2582 bp with 2 introns, respectively, and share sequence identities of 51.5–84.5% and 45.2–83.1% with homologs of other filamentous fungi. As illustrated in Figure 1A, the Set2 homologs share five domains known as SET, AWS (associated with SET domains), PostSET (cycteine-rich motif following a subdomain of SET domains), SRI (Set2 Rpb1 interacting), and WW (domain with two Trp/W residues), which is not predictable from the Set2 sequence of *P**. oryzae* among four fungi examined. The Ash1 homologs share the major SET domain and one or two associated PostSET domains. An NLS motif predicted with high marks (5.5–17) appears in the Set2 and Ash1 homologs of all examined fungi, suggesting a possible localization of each in nucleus.

The nuclear localization of either Set2 or Ash1 implicated by predicted NLS motif was well confirmed by subcellular localization of GFP-tagged Set2 or Ash1 fusion protein expressed in the WT strain, as shown in LSCM images (Figure 1B). Next, nuclear protein extracts isolated from the 3-day-old SDBY cultures of the Δ*set2* and Δ*ash1* mutants and their control strains were probed in the western blot experiments with appropriate antibodies (Appendix A) to characterize the catalytic activities of Set2 and Ash1 in not only H3K36me but also H3K4me and H3K9me. As a consequence, the mutants displayed the same signal levels of H3 accumulation in the nuclei as did the control strains but differentially attenuated signals of methylated lysine residues (Figure 1C). The H3me/H3 ratios demonstrated the signals of K36me2 and K36me3 attenuated by 89% and 67% in Δ*set2*, and of K36me1, K36me2 and K36me3 attenuated by 46%, 24% and 12% in Δ*ash1* relative to WT, respectively (Figure 1D). Unexpectedly, all signals of K4me1, K4Me2 and K4me3 were largely attenuated in the two mutants versus the WT strain, namely 49%, 54% and 72% attenuated in Δ*set2* and 88%, 71% and 92% attenuated in Δ*ash1*, respectively. Additionally, K9me2 was attenuated by 51% only in Δ*set2*. These data highlighted not only conserved activities of Set2 and Ash1 to K36me but also their noncanonical activities to K4me as well as the Set2’s activity to K9me2.

Interestingly, both Set2 and Ash1 showed greater roles in the catalysis of H3K4me3 than of H3K36me3, suggesting their catalytic activities closer to those of Set1/KMT2 elucidated previously in *B. bassiana* and *M. roberstii* [36,39]. The previous studies revealed a role of the KMT2-Cre1-Hyd4 pathway in mediating the *M. robertsii* pathogenesis via H3K4me3 as an epigenetic mark of *cre1* [36] and a similar role of the Set1-Cre1-Hyd1/2 pathway in the host infection, aerial conidiation and conidial hydrophobicity/adherence of *B. bassiana* [40]. To verify whether the Set2- and Ash1-reliant H3K4me3 involved in the Set1/KMT2-cored pathway, we assessed transcript levels of *cre1*, *hyd1*, *hyd2* and three other genes encoding hydrophobin-like proteins but functionally unknown as of yet. As a result, the expression of *cre1* was repressed by 88% in Δ*set2* and 84% in Δ*ash1* relative to the WT standard. Either *hyd1* or *hyd2* expression was markedly repressed in Δ*set2* and abolished in Δ*ash1*, accompanied by differential expressions of three other *hyd*-like genes.

Both catalytic activities and key gene transcripts altered in the absence of *set2* or *ash1* were well restored to the WT levels by targeted gene complementation. The results implicated involvements of Set2 and Ash1 in the Set1/KMT2-cored pathway to regulate in *B. bassiana* the expression of *cre1* essential for hyphal growth and development [51,52] and of both *hyd1* and *hyd2* required for hydrophobin biosynthesis and assembly into conidial surfaces [37].

### 3.2. Differential Roles of Set2 and Ash1 in Hyphal Growth, Conidiation and Conidial Quality

Compared to control strains, the Δ*set2* mutant showed moderate but significant defects (less than 15% reductions in colony diameter) in radial growth on minimal CDA or CDAs amended with some carbon or nitrogen sources under normal culture conditions, and its growth was more compromised (colony diameters diminished by 22–35%) on the carbon source of sodium acetate (NaAc) and the nitrogen source of NH_4_Cl or NH_4_NO_3_ and in the absence of carbon source but little defect on rich SDAY (Figure 2A). In contrast, the Δ*ash1* growth was facilitated on SDAY and exhibited insignificant or mitigated defects (colony diameters diminished by less than 10%) on the minimal media tested. In the stress assays, the Δ*set2* mutant was more sensitive to hyperosmotic, oxidative and cell wall perturbing stresses than the Δ*ash1* mutant, which even showed null response to NaCl (osmotic salt) and Congo red (cell wall stressor) in comparison to the control strains (Figure 2B). The estimates of relative growth inhibition revealed the Δ*set2* mutant’s hypersensitivity to oxidative stress induced by menadione (0.02 mM) or H_2_O_2_ (2 mM) and cell wall stress induced by Congo red (3 μg/mL) or calcofluor white (10 μg/mL) in CDA (Figure 2C).

Moreover, conidial yields were measured from the SDAY cultures during a 9-day incubation after 100 μL of a 10^7^ conidia/mL suspension was spread for culture initiation at the optimal regime. Compared to the WT strain, the Δ*set2* mutant suffered a reduction in conidial yield by 69% on day 5, 46% on day 7 and 42% on day 9, and the reductions greatly increased to ~98% in the Δ*ash1* cultures on days 5–9, although biomass accumulation level was unaffected in the Δ*set2* cultures and significantly increased in the Δ*ash1* cultures (Figure 2D). Aside from the severe conidiation defect, more indices of conidial quality were compromised in the absence of *ash1* than of *set2*, including markedly prolonged GT_50_, decreased UVB resistance, and lowered hydrophobicity, which was the sole index compromised in the absence of *set2* (Figure 2E).

Next, transcript levels of 32 genes likely associated with major changes of the examined phenotypes in the *set2* and *ash1* mutants were analyzed in the qPCR experiments. First, expression levels of the key CDP activator genes *brlA* and *abaA* in the 3-, 4- and 5-day-old SDAY cultures of the Δ*ash1* mutant relative to the WT strain were consistently reduced to hardly detectable levels (Figure 3A), coinciding well with its severe conidiation defect since conidiation was abolished in the absence of *brlA* or *abaA* [45]. In Δ*set2*, mitigated conidiation defect correlated with repressed expression of *abaA* instead of *brlA*. Second, 3 SOD genes were differentially expressed in the 3-day-old cultures of both Δ*set2* and Δ*ash1* (Figure 3B), including more repression in Δ*set2* than in Δ*ash1* of *sod2* encoding cytosolic MnSOD as a major contributor to total SOD activity [47]. Further shown in Figure 3B, two catalase genes (*cat1/catB* and *cat5/catP*) crucial for total catalase activity [48] were greatly downregulated in Δ*set2*, contrasting with less, but still significant, downregulation of only *cat2* in Δ*ash1*. As a result, Total SOD and catalase activities (Figure 3C) quantified in the protein extracts of the cultures were reduced by 13% and 95% in Δ*set2*, and 5.3% (*p* < 0.05 in Tukey’s HSD test) and 59% in Δ*ash1*, respectively. Third, more MAPK Slt2- and Hog1-cascaded kinase genes required for cell wall integrity and osmoregulation [49,50] were repressed at the significance of one-fold transcript change in Δ*set2* (*slt2*, *mkk1*, *bck1*, *pbs2* and *sskB*) than in Δ*ash1* (*slt2*, *bck1* and *sskB*), and the same was true for other 11 genes putatively involved in biosynthesis and assembly of cell wall components (Figure 3D).

All of the above phenotypic, transcriptional and enzymatic changes observed in the Δ*set2* and Δ*ash1* mutants were well restored in the corresponding complementation mutants. These data indicated greater role of Set2 than of Ash1 in the utilization of carbon and nitrogen sources for hyphal growth and cellular responses to different types of stress cues but of Ash1 than of Set2 in aerial conidiation crucial for the asexual cycle in vitro of *B. bassiana*.

### 3.3. Differential Roles of Set2 and Ash1 in Virulence-Related Cellular Events

In the standardized bioassays, *G. mellonella* larvae infected by the Δ*set2* and Δ*ash1* mutants survived longer than those infected by the control strains in either mode (Figure 4A). LT_50_s estimated by modeling analysis of time-course mortality trends were averagely prolonged by 25% and 21% for the 2 mutants against the model insect via NCI and 42% and 36% via CBI, respectively, in comparison to the mean (±SD) LT_50_ estimates of 5.5 (±0.09) and 3.5 (±0.07) days for all control strains (Figure 4B). Thus, the fungal virulence via NCI or CBI was slightly more attenuated in the absence of *set2* than of *ash1*.

For insight into the attenuated virulence via NCI, conidial adherence to insect cuticle assayed on locust hind wings (Figure 4C) decreased by 53.6% in Δ*set2* and 40.0% in Δ*ash1* relative to the control strains (Figure 4D). The incubation of mycosis-killed larvae at 25 °C resulted in heavier or much heavier outgrowths of all control strains on the surfaces of cadavers 5 days post-death than those formed by Δ*ash1* or Δ*set2* (Figure 4E). The difference in cadaver outgrowths between the two mutants suggested that the Δ*set2* mutant grew out of the cadavers slower than the Δ*ash1* mutant, as observed on the scant media. Next, the total activities of ECEs measured from the supernatants of 3-day-old CDB-BSA cultures were reduced by 44% and 72% in Δ*set2*and Δ*ash1* in comparison to their control strains, but the activities of Pr1 proteases which are collectively important for cuticle penetration [43] showed no variation (*F*_4,10_ = 0.35, *p* = 0.84) among the tested strains (Figure 4F). Biomass levels in the cultures were unaffected for Δ*set2* (*p* > 0.05 in Tukey’s HSD test) but markedly increased by 70% for Δ*ash1*, implicating a more reduced secretion of ECEs by the cultures of Δ*ash1* than of Δ*set2*. A status of hemocoel colonization by the control strains was featured by the presence of abundant hyphal bodies in the hemolymph samples taken from the larvae surviving 72 h post-injection (hpi) while the hyphal bodies of the Δ*set2* and Δ*ash1* mutants were hardly observed in the samples until 96 hpi (Figure 4G). Consequently, concentrations of hyphal bodies measured from the hemolymph samples were averagely reduced by 74% and 72% for the two mutants at 72 hpi, and the reductions diminished to 66% and 42% at 96 hpi, respectively, in comparison to the measurements from the control strains (Figure 4H). Illustrated in Figure 4I, the 3-day incubation of conidia in TPB mimicking insect hemolymph resulted in 20% and 37% reductions in dimorphic (hypha-blastospore) transition rates in the Δ*set2* and Δ*ash1*cultures, respectively, and greater difference in biomass accumulation between the two mutants (10% decrease versus 32% increase).

These data and observations demonstrated that, in *B. bassiana*, Set2 and Ash1 played important, but differential, roles in sustaining the virulence-related cellular events critical for conidial adherence, cuticle penetration, hemocoel colonization, and proliferation in vivo by yeast-like budding to speed up host death from mummification. The Δ*set2* mutant was more compromised in conidial adherence to insect cuticle and proliferation in vivo than the Δ*ash1* mutant, and vice versa in the secretion of some cuticle-degrading enzymes not including Pr1 proteases.

## 4. Discussion

As typical KMT3 enzymes, Set2 and Ash1 were proven to have both conserved catalytic activities to H3K36me and noncanonical activities to H3K4me as well as the Set2’s activity to H3K9me2 in *B. bassiana*. Previously, noncanonical catalytic activities were displayed by H3K9me-specific Dim5/KMT1 to H3K4me1/me2 and H3K36me2 [40] and also by H3K4me-specific Set1/KMT2 to H3K36me2 [39] when nuclear protein extracts were probed with specific anti-methyl antibodies (Appendix A). The previous and present studies unravel that the KMT1, KMT2 and KMT3 enzymes could have differentiated in catalytic and transcriptional activities to support insect-pathogenic lifestyle of *B. bassiana* by making use of nutrients from the broadest host spectrum. Such a differentiation could have also occurred in many other insect and plant mycopathogens that adapt to broad or specific host spectra. However, methodology is critical for probing effectively all catalytic activities of a H3 lysine-specific KMT. In our experience, the use of nuclear protein extracts in western experiments is superior to the use of total protein extracts that may contain insufficient or weak signal from nuclei. This is because H3me occurs exclusively in the nuclei. Interestingly, broader/greater roles of Set2 and Ash1 in the catalysis of H3K4me than of H3K36me suggest that both of them function like Set1 and hence involve in sequential activation of *cre1* required for carbon catabolite repression in association with the utilization of insect nutrients and of *hyd1* and *hyd2* essential for conidial hydrophobicity and infection-required adherence to insect cuticle, as discussed below.

In the present study, noncanonical H3K4me3 was more attenuated in Δ*ash1* (92%) than in Δ*set2* (72%), contrasting with conserved H3K36me3 attenuated by 12% and 67% in the two mutants, respectively. Previously, KMT2-dependent H3K4me3 was characterized as an epigenetic mark of *cre1* for its transcriptional activation leading to upregulation of *hyd4* in *M. robertsii*, unveiling a regulatory role of the KMT2-Cre1-Hyd4 pathway in the fungal pathogenesis [36]. Since Hyd4 in *M. roberstii* is homologous to both *hyd1* and *hyd2* elucidated as regulators of hydrophobin biosynthesis and assembly into an outermost rodlet-bundle layer of conidial coat in *B. bassiana* [37,39], a similar Set1- Cre1-Hyd1/2 pathway was revealed to regulate conidial hydrophobicity and adherence to insect cuticle, virulence via NCI or CBI, and asexual development [39]. The H3K4me3 levels detected in Δ*set2* and Δ*ash1* were close or even identical to those seen in the previous Δ*kmt2* and Δ*set1* mutants, implicating that Set2 and Ash2 could function like Set1 in *B. bassiana*. This implication was verified by greatly repressed or nearly abolished expressions of *cre1*, *hyd1* and *hyd2* and also by reduced conidial hydrophobicity and adherence to insect cuticle in Δ*set2* and Δ*ash1*, suggesting a possibility for either Set2 or Ash1 to play alternative roles in the Set1-cored pathway. The possibility of Ash1 is greater than of Set2 since H3K4me3 was nearly abolished in Δ*ash1* as seen in the previous Δ*kmt2* and Δ*set1* mutants. We infer that the involvements of more KMTs in the mediation of *hyd1* and *hyd2* via transcriptional activation of *cre1* marked by H3K4me3 could increase conidial hydrophobicity and adherence to insect cuticle, enhance fungal capability of utilizing scant nutrients in insect integument for entry into host hemocoel, where hemolymph nutrients are rich, and hence favor fungal adaptation to insect-pathogenic lifestyle. From this point, the roles of Set1, Set2 and Ash1 in the co-catalysis of H3K4me3 for sequential mediation of *cre1* and key *hyd* genes could be an important mechanism underlying the *B. bassiana*’s adaptation to the broadest host spectrum. Thus, the fungal virulence was inevitably attenuated via NCI or CBI when *set2* or *ash1* lost function.

Aside from differential roles in the upregulation of *hyd1* and *hyd2*, Ash1 played greater role than Set2 in sustaining conidiation capacity by transcriptional activation of *brlA* and *abaA* as key CDP activator genes. In *B. bassiana*, either aerial conidiation in plate cultures or submerged blastospore formation (dimorphic transition) in TPB cultures mimicking a situation in insect hemolymph was completely abolished in the absence of *brlA* or *abaA* [45]. In Δ*ash1*, extremely severe conidiation defects correlated well with nearly abolished expression of both *brlA* and *abaA* in the SDAY cultures with increased biomass accumulation, while mitigated conidiation defects correlated with repressed expression of only *abaA* in the Δ*set2* cultures with unaffected biomass accumulation. Interestingly, the dimorphic transition rate was higher in the TPB cultures of Δ*set2* than of Δ*ash1* although the latter mutant had accumulated much more biomass in the cultures. This phenomenon is consistent with more repression of *brlA* and *abaA* in Δ*ash1* but conflicting with faster increase in its hyphal bodies in insect hemolymph post-injection, suggesting some other factors involved in the course of proliferation in vivo. Previously, the injection of conidia into insect hemocoel induced the aggregation of host hemocytes and the encapsulation of injected conidia by aggregated hemocytes during the first 48 h period of germination and growth, leading to hyphal bodies hardly observed until 72 h after injection [53]. Breaking the encapsulation of fungal cells by aggregated host hemocytes relies upon fungal antioxidant activity to scavenge reactive oxygen species, such as superoxide anions and H_2_O_2_, from host immune defense [2]. In the present study, the Δ*set2* mutant exhibited higher sensitivity than the Δ*ash1* mutant to oxidative stress induced by menadione (superoxide anions-generating compound) or H_2_O_2_, accompanied by repressed expression of more key antioxidant enzyme genes and more reductions in total SOD and catalase activities required for the respective decomposition of superoxide anions and H_2_O_2_ [47,48]. Moreover, more kinase genes in the signaling Hog1 and Slt2 MAPK cascades, which have proved to interplay and regulate multiples stress responses in *B. bassiana* [49,50], were significantly downregulated in Δ*set2* than in Δ*ash1*. Taking these results into account, it is not difficult to infer that the Δ*set2* mutant could take longer time than the Δ*ash1* mutant to collapse host immune defense for the release of hyphal bodies into host hemocoel, where they proliferate by yeast-like budding until host mummification to death, and hence was slightly more compromised in virulence via NCI or CBI. In other words, Set2 plays more important roles than does Ash1 in transcriptional mediation of stress-responsive signaling and effector genes involved in cellular responses to stress cues encountered inside and outside host.

Overall, the main KMT1, KMT2 and KMT3 enzymes characterized in the present and previous studies [39,40] play important, but differential, roles in orchestrating cellular processes and events associated with *B. bassiana*’s host infection, pathogenesis, virulence, and conidiation required for survival/dispersal in host habitats, as seen in the plant-pathogenic fungi *B. cinerea* [26], *F. verticillioides* [27,28,31], *F.*
*fujikuroi* [29,32] and *M. oryzae* [30,33]. Notably, all of the ‘H3 lysine-specific’ KMTs have not only conserved but also noncanonical catalytic activities in *B. bassiana*. Particularly, both Set2 and Ash1 have even greater roles in the catalysis of noncanocial H3K4me3 than of conserved H3K36me3, offering a novel insight into the regulatory roles of Set2 and Ash1 in transcriptional activation of *cre1* and key *hyd* genes as that of Set1/KMT2 elucidated previously [36,39]. Nonetheless, previous studies paid little attention to noncanonical activities of plant- pathogenic fungal KMTs except Set1 in *A. flavus* [34]. Among mono-, di- and trimethylated signals of H3K4, H3K9 and H3K36 residues in filamentous fungi, only H3K4me3 mediated by Set1 has proved to be an epigenetic mark of *cre1* for its activation leading to upregulation of key *hyd* gene in *M. robertsii* and of *hyd1* and *hyd2* in *B. bassiana*. It remains a great challenge to identify the targets of all epigenetic marks depending on conserved and noncanonical activities of KMT1, KMT2 and KMT3 enzymes, warranting future studies for in-depth insight into epigenetic mechanisms underlying their pleiotropic effects on filamentous fungal lifestyles.

## Figures and Tables

**Figure 1 jof-07-00956-f001:**
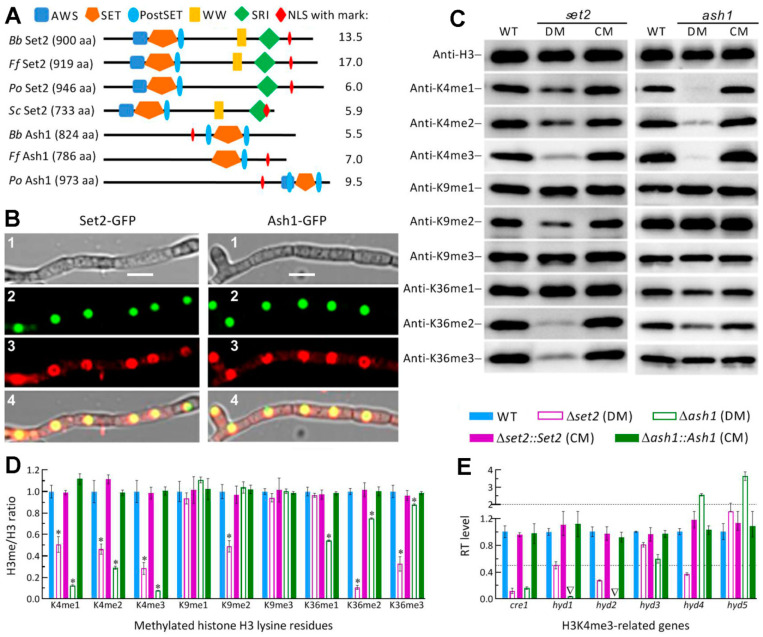
Subcellular localization and catalytic activities of Set2 and Ash1 in *B. bassiana*. (**A**) Main domains and NLS motif with mark predicted from the Set2 and Ash1 homologs of *B. bassiana* (*Bb*), *Fusarium*
*fujikuroi* (*Ff*), *Pyricularia*
*oryzae* (*Po*) and *Saccharomyces cerevisiae* (*Sc*) at http://smart.embl-heidelberg.de/ and http://nls-mapper.iab.keio.ac.jp/, respectively. (**B**) LSCM images (scale: 5 μm) for subcellular localization of Set2-GFP and Ash1-GFP fusion proteins in the hyphal cells stained with the nuclear dye DAPI (shown in red) after collection from a 48-h-old SDBY culture grown on a shaking bed at 25 °C. Panels 1, 2, 3 and 4 are bright, expressed, stained, and merged views of the same field. (**C**) Western blots for the signals of H3K4me1/me2/m3, H3K9me1/me2/me3 and H3K36me1/me2/me3 in the nuclear protein extracts isolated from the 3-day-old SDBY cultures of the WT, Δ*set2* (DM), Δ*set2**::**Set2* (CM), Δ*ash1* (DM) and Δ*ash1**::**Ash1* (CM) strains. Aliquots of 40 μg protein extracts were probed with appropriate antibodies (detailed in Appendix A). (**D**) Signal intensity ratios of methylated H3 lysine residues versus nuclear H3 accumulation (H3me/H3 ratio) quantified from the blots of three proteins samples per strain. * *p* < 0.001 for all ratios except the K36me3/H3 ratio of Δ*ash1 (p* < 0.05) in Tukey’s HSD tests. (**E**) Relative transcript (RT) levels of H3K4me3-related genes in the *set2* and *ash1* mutants with respect to the WT strain. The upper and lower dashed lines denote significant levels of one-fold up- and downregulation. Inverted triangle indicates abolished gene expression. Error bars: Standard deviations (SDs) of the means from three independent replicates.

**Figure 2 jof-07-00956-f002:**
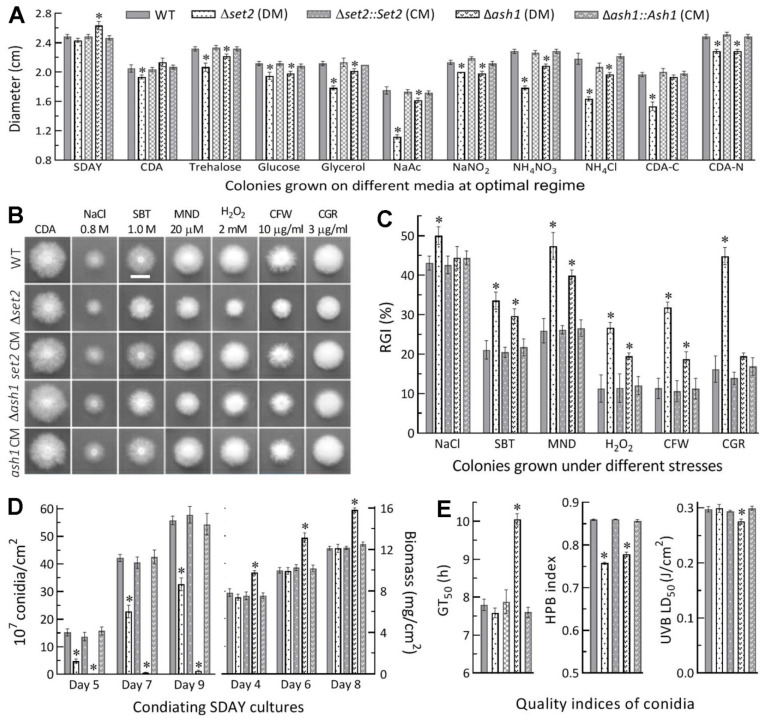
Differential roles of *set2* and *ash1* in sustaining radial growth, stress tolerance, conidiation and conidial quality of *B. bassiana*. (**A**) Diameters of 8-day-old colonies grown at 25 °C on rich SDAY, minimal CDA and CDAs amended with different carbon or nitrogen sources and with deleted carbon or nitrogen source. (**B**,**C**) Images (scale: 10 mm) of fungal colonies grown at 25 °C for 8 days on CAD alone (control) or supplemented with the indicated concentrations of chemical stressors (MND, menadione; SBT, sorbitol; CGR, Congo red; CFW, calcofluor white) and relative growth inhibition (RGI) percentages of fungal strains under the stresses. All colonies were initiated by spotting 1 μL aliquots of a 10^6^ conidia/mL suspension. (**D**) Conidial yields and biomass levels quantified from the SDAY cultures during a 9-day incubation after the cultures were initiated by spreading 100 μL aliquots of a 10^6^ conidia/mL suspension at the optimal regime of 25 °C and L:D 12:12. (**E**) The indices of conidial quality assessed as the estimates of GT_50_ (h) for 50% of conidial germination at 25 °C, hydrophobicity (HPB) in the aqueous-organic system, and LD_50_ (J/cm^2^) for resistance to UVB irradiation. * *p* < 0.05 in Tukey’s HSD tests. Error bars: SDs of the means from three independent replicates.

**Figure 3 jof-07-00956-f003:**
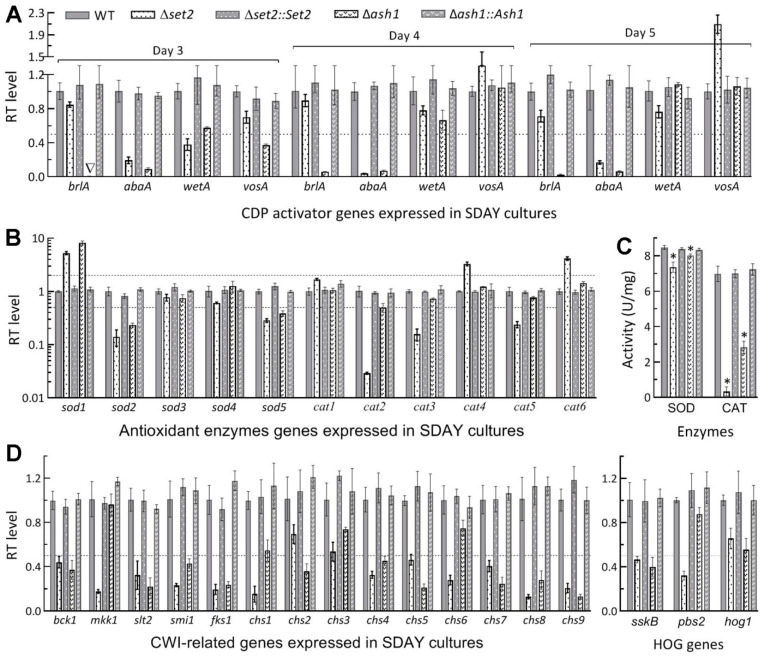
Subsets of genes differentially repressed in the *set2* and *ash1* mutants relative to the WT strain. (**A**) Relative transcript (RT) levels of 3 CDP activator genes and downstream *vosA* in 3-, 4- and 5-day-old SDAY cultures grown at the optimal regime of 25 °C and L:D 12:12. (**B**,**C**) RT levels of 11 antioxidant enzyme genes and total SOD and catalase (CAT) activities in the 3-day-old SDAY cultures. (**D**) RT levels of MAPK Slt2- and Hog1-cascaded kinase genes and those involved in cell wall integrity (CWI). Dashed line indicates a significance of one-fold downregulation. * *p* < 0.05 in Tukey’s HSD tests. Error bars: SDs of the means from three independent cDNA samples (**A**,**B**,**D**) or protein extracts (**C**).

**Figure 4 jof-07-00956-f004:**
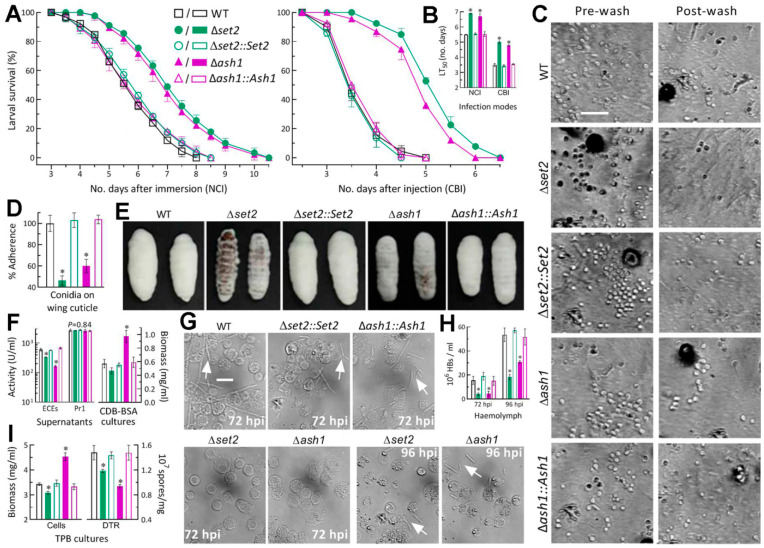
Differential roles of *set2* and *ash1* in host infection- and virulence-related cellular events of *B. bassiana*. (**A**) Survival trends of *G.*
*mellonella* larvae after topical application (immersion) of a 10^7^ conidia/mL suspension for normal cuticle infection (NCI) and intrahemocoel injection of ~500 conidia per larva for cuticle-bypassing infection (CBI). (**B**) LT_50_ (no. days) estimates made by modeling analysis of time-mortality trends. (**C**,**D**) Images (scale: 20 μm) and estimates for conidial adherence to locust hind wing cuticle. The estimates are percent ratios of post-wash counts over pre-wash counts with respect to the WT standard. (**E**) Images of hyphal outgrowths on the surfaces of insect cadavers 5 days post-death. (**F**) Biomass accumulation levels and total activities of cuticle-degrading extracellular enzymes (ECEs) and Pr1 family proteases quantified from the 3-day-old CDB-BSA cultures prepared by shaking 10^6^ conidia/mL suspensions at 25 °C. (**G**,**H**) Microscopic images (scale: 20 μm) and concentrations of hyphal bodies (HBs; arrowed) in hemolymph samples taken from the larvae surviving 72 and 96 h post-injection (hpi). (**I**) Biomass levels and dimorphic transition rates measured from the 3-day-old cultures incubated at 25 °C by shaking 10^6^ conidia/mL suspensions in TPB mimicking insect hemolymph. * *p* < 0.05 in Tukey’s HSD tests. Error bars: SDs of the means from three independent replicates.

## Data Availability

All data presented in this study are included in the paper and associated Appendix A.

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
