# Peer review of "Conserved and Noncanonical Activities of Two Histone H3K36 Methyltransferases Required for Insect-Pathogenic Lifestyle of *Beauveria bassiana"

_jof, 2021, doi:10.3390/jof7110956_

Round 1

Reviewer 1 Report

I have only one major concern about this topic:
For the “Transcriptional Profiling”, how to pick up 32 phenotype-related genes and how to prove their relationship to the mutations? I think there might have more widely genetic network in the mutations. Instead of pick the group of phenotype-related genes, RNA-seq is needed to evaluate for the network, which involve in the set2 and ash1 mutants 

The minor comments:
1)    Figure 1A: What does the number mean in the right size ? (13.5, 17.0, 6.0…) 
2)    Figure 1E doesn’ t have any legend to explain the data? And also what does the RT level mean? 
3)    Figure 2B please specifically indicate the “CM” for delta-set2 and delta-ash1
4)    In the conclusion, the author should provide more perspective for the application of their finding, i.e., how does the epigenetic regulation improve the fungi to kill the insect or more stress- tolerance, etc. 

Author Response

I have only one major concern about this topic:
For the “Transcriptional Profiling”, how to pick up 32 phenotype-related genes and how to prove their relationship to the mutations? I think there might have more widely genetic network in the mutations. Instead of pick the group of phenotype-related genes, RNA-seq is needed to evaluate for the network, which involve in the set2 and ash1 mutants 

Author response: Good questions! Selection of phenotype-related genes for transcriptional analysis is based on fully understanding functions of subsets of genes characterized in a studied fungus. Yes, transcriptomic analysis by RNA-seq may provide a genome-wide view of those genes differentially expressed in the absence of a target gene. Such a genome-wide effect has been elucidated in the absence of dim5/kmt1 required for H3K36me3 (Ren et al. 2021), of set1/kmt2 required for histone H3K4me3 (Lai et al. 2020) and of H3K4me3-regulated cre1 essential for carbon catabolite repression (Mohamed et al. 2021). In the present study, greater role of either set2 or ash1 in the catalysis of H3K4me3 than of H3K36me3 guided us to focus on their roles in the Set1-Cre1-Hyd1/2 pathway and other important phenotype-related genes found in the previous transcriptomes. Therefore, it is unnecessary to perform RNA-seq analysis in this study.

The minor comments:
1)    Figure 1A: What does the number mean in the right size ? (13.5, 17.0, 6.0…) 

Author response: The values denote predicted NLS marks, which has been clarified in this revision

2)    Figure 1E doesn’ t have any legend to explain the data? And also what does the RT level mean? 

Author response: The missed legend has been added to Fig. 1E.

3)    Figure 2B please specifically indicate the “CM” for delta-set2 and delta-ash1

Author response: Revised as suggested.

4)    In the conclusion, the author should provide more perspective for the application of their finding, i.e., how does the epigenetic regulation improve the fungi to kill the insect or more stress- tolerance, etc. 

Author response: Good point! It is a great challenge to clarify what genes are the targets of epigenetic marks represented by mono-, di- and trimethylated signals of all H3 lysine residues as conserved and noncanoical activities of a histone methyltransferase in filamentous fungi. To date, only H3K4me3 has been characterized as an epigenetic mark of the carbon catabolite repressor gene cre1. Large funding and labor inputs are needed to clarify the targets of those epigenetic marks for in-depth insight into overall epigenetic mechanisms in a fungal insect pathogen.

Reviewer 2 Report

The authors are presenting broader roles of histone methyltransferases, Set2 and Ash1 in insect pathogen B. bassiana. The manuscript is already in a good written form and provide important information on basic molecular biological/pathological fields in entmopathogenic fungi. I believe the manuscript merits publication in “Journal of Fungi” after minor revisions.

  • 1C; Please correct “Anti-K36me1” to “Anti-K36me2”.
  • 2E; Harmonize the Fig. and Fig. Legend.
  • Please present the source of Abs (anti-H3, anti-K4me, etc) used in Western Blot Assay.

Author Response

The authors are presenting broader roles of histone methyltransferases, Set2 and Ash1 in insect pathogen Bbassiana. The manuscript is already in a good written form and provide important information on basic molecular biological/pathological fields in entmopathogenic fungi. I believe the manuscript merits publication in “Journal of Fungi” after minor revisions.

Author response: Thanks a lot for understanding and encouragement.

  • 1C; Please correct “Anti-K36me1” to “Anti-K36me2”.

Author response: Revised as suggested.

  • 2E; Harmonize the Fig. and Fig. Legend.

Author response: Fig. 2E and legend have been harmonized.

  • Please present the source of Abs (anti-H3, anti-K4me, etc) used in Western Blot Assay.

Author response: Anti-H3 and nine anti-methyl antibodies used in the study are listed in Table S2.

Round 2

Reviewer 1 Report

I have no questions about this manuscript.